# You Told Me That Joke Twice: A Systematic Investigation of Transferability and Robustness of Humor Detection Models

**Alexander Baranov[1]** and **Vladimir Kniazhevsky[1]** and **Pavel Braslavski[1,2]**

[1]HSE University, Moscow, Russia

[2]School of Engineering and Digital Sciences, Nazarbayev University, Astana, Kazakhstan

`ambaranov@hse.ru, wlad.kniazhewski@gmail.com, pbras@yandex.ru`

## Abstract

In this study, we focus on automatic humor detection, a highly relevant task for conversational AI. To date, there are several English datasets for this task, but little research on how models trained on them generalize and behave in the wild. To fill this gap, we carefully analyze existing datasets, train RoBERTa-based and Naïve Bayes classifiers on each of them, and test on the rest. Training and testing on the same dataset yields good results, but the transferability of the models varies widely. Models trained on datasets with jokes from different sources show better transferability, while the amount of training data has a smaller impact. The behavior of the models on out-of-domain data is unstable, suggesting that some of the models overfit, while others learn non-specific humor characteristics. An adversarial attack shows that models trained on pun datasets are less robust. We also evaluate the sense of humor of the chatGPT and Flan-UL2 models in a zero-shot scenario. The LLMs demonstrate competitive results on humor datasets and a more stable behavior on out-of-domain data. We believe that the obtained results will facilitate the development of new datasets and evaluation methodologies in the field of computational humor. We've made all the data from the study and the trained models publicly available: https://github.com/Humor-Research/Humor-detection.

Tell me which are funny, which are not – and which get a giggle first time but are cold pancakes without honey to hear twice.

—Robert Heinlein, *The Moon Is a Harsh Mistress*

## 1 Introduction

In Robert Heinlein's science fiction novel, the main character, Mannie, uses a collection of jokes to track the development of self-awareness in a supercomputer nicknamed Mike. This reflects a common belief that a sense of humor is an innate human trait.

Since Alan Turing proposed his *imitation game* (Turing, 1950), a computer's ability to carry on a conversation has been one of the criteria of its intelligence. Clark et al. (2019) conducted a series of interviews to understand what people expect from conversations with artificial agents, and what characteristics they consider important in conversations in general. Participants mentioned humor as an important attribute that adds substance to discussions and can be a key driver of conversations. They also described humor as a desirable feature that can make conversations with agents more engaging and entertaining.

Humor theories can be traced back to Aristotle and have been elaborated by various disciplines, including semantics, psychology, and linguistics (Raskin, 1984; Attardo, 1994; Roeckelein, 2002). The development of conversational AI makes computational humor methods highly relevant and in demand for practical applications. However, the difficulty of developing automatic methods in this area is determined by humor properties such as diversity and unpredictability. Humor is an umbrella term for many related yet distinct phenomena. In this study, we focus on *verbal* humor, but it can be still quite diverse. For example, *puns* are based on wordplay, while *satire* refers to real-life contexts, *irony* may appear serious at first glance, *canned* jokes have a relatively stable form and circulate among a wide audience, unlike *spontaneous* jokes, which are improvised and context-dependent.

To date, there are several English datasets for binary humor detection task that vary widely in size and sources of humorous content. However, there is little research on how models trained on them generalize and behave in the wild. To fill this gap, we carefully collected nine datasets suitable for humor detection, analyzed and cleaned them, and formatted uniformly. We fine-tuned the RoBERTa models on each of the datasets and tested them on

the rest. We also compared the results with those of Naïve Bayes (NB) classifiers. In addition, we evaluated the humor capabilities of two LLMs: chatGPT and Flan-UL2. We also conducted an adversarial attack and applied the models to supplementary collections of figurative language, ironic tweets, and utterances from fiction and TV series.

Our analysis confirms that standard RoBERTa-based models overfit on particular training sets. On several datasets, the Transformer-based models don't improve over the unigram-based NB classifier. Based on the obtained results, we can conclude that the crucial factor for a better generalization is the diversity of humor types in the training, not the size of the dataset. This is confirmed by the experiments on a medium-sized collection compiled from different sources. All but three models are quite resistant to an adversarial attack based on word substitution. The humor detection capabilities of the LLMs are quite competitive across different datasets, although don't show superior results on individual collections; these results are not based on memorization of jokes. The behavior of the models on supplementary collections is mixed. In general, the models assign texts to the humorous class more often than we expected, which may indicate overfitting. Some models violate our assumed behavior based on the genres of the novels and TV series. The LLMs show a more stable behavior on supplementary collections, consistent with our expected behavior of a 'good humor detector'.

Based on the results obtained, we can conclude that humor is a complex and multifaceted phenomenon, and the task of humor detection must try to reflect this complexity. We have shown that one way to improve the generalizability of humor detection models is to train them on diverse collections. Conversational systems are probably the most promising application for humor detection. Accordingly, it is desirable to build new humor-related datasets that reflect the peculiarities of this application domain, taking into account different dialog scenarios and user characteristics.

We believe that the results obtained will facilitate the development of new datasets and evaluation methods in the field of computational humor. We have made the datasets in a unified format and the trained models freely available.[1]

---

[1] https://github.com/Humor-Research/Humor-detection

## 2 Related Work

An early work on automatic humor detection was specifically targeted at wordplay in 'knock-knock' jokes (Taylor and Mazlack, 2004). Mihalcea and Strapparava (2005) addressed the humor detection task in a broader context and automatically compiled a large and diverse dataset of humorous and non-humorous short texts for training and testing humor classifiers. They conducted extensive experiments with different humor-specific features such as alliteration, rhyme, antonymy and adult slang, as well as lexical features using Naïve Bayes and SVM classifiers. Yang et al. (2015) introduced the notion of 'humor anchors' – lexical units that enable humorous effect. They also addressed humor detection employing Gradient Boosting Regression Trees with a wide variety of features, including humor-specific features and *word2vec* embeddings. Liu et al. (2018) demonstrated utility of syntax features for humor detection. Abulaish et al. (2020) survey and summarize different features used in humor classification studies. There are studies concerned with humor detection in specific domains, such as Twitter (Zhang and Liu, 2014) or product question answering systems (Ziser et al., 2020).

With the advent of neural networks, computational humor research has shifted from feature engineering to experiments with different neural architectures on larger collections. Chen and Soo (2018) applied Convolutional Neural Network (CNN) to humor detection. Weller and Seppi (2019) demonstrated that the BERT-based classifier achieves superior results in humor recognition task on several existing datasets, and also proposed a new dataset composed of humorous Reddit posts. There are studies leveraging the set-up–punchline structure of jokes for humor detection task using BERT senetence embeddings (Annamoradnejad and Zoghi, 2020) or GPT-2 (Xie et al., 2021). Peyrard et al. (2021) applied various models – three transformer models (BERT, distilBERT, and RoBERTa), fastText-based representation, GPT-2, and LSTM, – to a collection of aligned humorous/non-humorous sentence pairs. The authors discovered that humor classification occurs in the last transformer layers meaning that the model relies on semantic, rather than lexical features, and one of the transformer heads specialized in attending to the humorous part of the input text. This study is complement to ours: we don't perform probing and extensive comparison over

different models, but pay the main attention to *one* transformer-based model and *various* datasets. In a concurrent paper, Arora et al. (2022) improved humor detection by using several datasets containing different humor species at once, although didn't investigate generalization of models trained on individual datasets.

Several humor-related shared tasks organized in recent years have contributed to the NLP community's interest in computational humor and facilitated progress in the field (Miller et al., 2017; Potash et al., 2017; Van Hee et al., 2018; Castro et al., 2018a; Hossain et al., 2020a; Meaney et al., 2021). In this paper, we focus on verbal humor, but there are multimodal humor detection studies leveraging images, video, or audio along with texts (Radev et al., 2016; Shahaf et al., 2015; Hasan et al., 2019; Bertero and Fung, 2016). Besides English data, there are humor-related datasets for Italian (Buscaldi and Rosso, 2007), Spanish (Castro et al., 2018b), Dutch (Winters and Delobelle, 2020), Russian (Blinov et al., 2019), and Chinese (Zhang et al., 2019).

Modern large language models (LLMs) are impressive in their ability to engage in meaningful, context-aware conversation, answer questions, generate text, and program code. However, as a systematic evaluation showed, LLMs tend to perform worse than specialized models at certain tasks (Kocoń et al., 2023). Borji (2023) reports that chatGPT 'has some understanding of humor', but provides examples of failure and points out the need for a comprehensive examination of the humor capabilities of LLMs. Goes et al. (2023) elaborated a set of prompts to evaluate the GPT-4's ability to judge the funniness of jokes; one of the variants showed 'a weak but encouraging positive correlation with human judges.' A recent study investigated chatGPT's ability to generate and interpret jokes (Jentzsch and Kersting, 2023). The authors found out that chatGPT's humorous repertoire was quite limited and identified 25 top jokes.

## 3 Data

### 3.1 Humor datasets

We have collected nine datasets suitable for humor detection that have been presented in previous studies (one of the datasets is a merge of two 'sibling' datasets). We supplemented the set with a collection of headlines from satirical newspaper *The Onion*. We have also created a combined dataset

from existing ones to ensure the diversity of humorous texts, which may improve the transferability of models trained on the data. Table 6 in the Appendix lists the datasets, reference papers, download URLs, and licenses where available.

The dataset collected by Mihalcea and Strapparava (2005) has been a *de facto* standard for humor recognition studies for years. It contains 16,000 one-liners collected online (hence its name, **16kOL**[2] for short) and 16,000 non-humorous sentences from news titles, proverbs, British National Corpus, and Open Mind Common Sense collection. Manual assessment of a small sample performed by the authors estimates the level of potential noise in the dataset to be around 9%.

Another dataset used in several studies comprises of 2,400 puns from the website 'Pun of the day' (**PotD**) and an equal number of negative samples from the news, Yahoo!Answers, and collections of proverbs (Yang et al., 2015). EnglishPuns (**EnPuns**) contains about 4,000 short texts from various sources – puns, non-punny jokes, aphorisms, etc., with puns being the positive class (Miller et al., 2017). This dataset has an additional level of annotation: pun-enabling words are annotated with their WordNet senses; this annotation is not used in the current study.

ShortJokes (**ShJ**) dataset combines jokes scraped from online collections and humorous Reddit posts. Chen and Soo (2018) complemented this existing collection with 'serious' part sourced from a news collection and used the dataset for their humor classification experiments. Later, Weller and Seppi (2019) reproduced this complement and made the dataset publicly available.[3] **ShJ** is by far the largest dataset for humor classification. Hahackathon (**Haha**) dataset was used in the SemEval 2021 task 'Detecting and Rating Humor and Offense' (Meaney et al., 2021). The data was sourced from Twitter (80%) and sampled from the **ShJ** dataset (20%) using a list of keywords potentially signalling offensive content and subsequently annotated manually. Weller and Seppi (2019) collected jokes from Reddit (**ReJ**) and split them into less and more funny ones based on users' upvotes, and then balanced the classes for training and test-

---

[2]In the rest of the paper, we use abbreviations to refer to both the datasets and the models trained on them; the usage will be clear from the context.

[3]The dataset was published without labels for the test subset; we restored them by matching the instances to the original humorous part.

| Dataset | ☺ | $N^+$ | $N^-$ | %\$#$^+$ | %\$#$^-$ | $l^+$ | $l^-$ | $KL$ | train / dev / test |
|---|---|---|---|---|---|---|---|---|---|
| 16kOL | Web | 15,979 | 15,717 | 491 | 114 | 14.77 | 10.34 | 1.70 | 22,188 / 3,170 / 6,338 |
| PotD | Web | 2,423 | 2,323 | 10 | 13 | 13.22 | 13.80 | 1.31 | 3,323 / 475 / 948 |
| EnPuns | Web | 2,875 | 1,152 | 15 | 11 | 14.18 | 10.72 | 2.28 | 2,819 / 403 / 805 |
| ShJ | Web/Re | 232,137 | 234,054 | 46,629 | 6,457 | 18.92 | 21.26 | 2.29 | 347,486 / 57,914 / 60,791* |
| ReJ | Re | 10,327 | 10,327 | 2,554 | 1,468 | 95.83 | 40.88 | 0.95 | 19,438 / 608 / 608* |
| Haha | Twi/Re | 6,179 | 3,821 | 701 | 94 | 24.18 | 26.07 | 1.48 | 8,000 / 1,000 / 1,000* |
| FL+HME | Editing | 13,432 | 10,165 | 350 | 200 | 12.54 | 12.57 | 0.30 | 16,518 / 2,360 / 4,719 |
| Unfun.me | Oni | 821 | 969 | 38 | 15 | 10.02 | 9.11 | 0.68 | 1253 / 179 / 358 |
| NF | Re | 88,089 | 10,710 | 19,722 | 1,073 | 27.37 | 778.86 | 4.13 | 69,160 / 9,880 / 19,759 |
| TheO | Oni | 8,952 | – | 735 | – | 15.18 | – | – | 6,284 / 885 / 1,783 |
| COMB | Web/Re/Oni | 28,287 | 17,201 | 4,668 | 700 | 18.11 | 20.54 | 1.20 | 28,860 / 7,539 / 9,089 |

Table 1: Humor detection datasets used in the study and their statistics (abbreviations are deciphered in the text). Source of funny part (☺): *Web* (collections), *Re*(ddit), *Twi*(tter), (human) *Editing*, (The)*Oni*(on) satire news. $N$ – size of positive (funny) and negative (serious) parts; %\$# – number of texts containing obscene words in positive/negative class, respectively; $l$ – average text length in words in positive/negative class; $KL$ – symmetrized smoothed Kullback-Leibler divergence between word distributions in positive and negative classes; * marks datasets with original *train / dev / test* splits.

ing. Interestingly, humorousness scores obtained from Reddit users' votes and by crowd workers are quite different. Note that **ReJ** and **EnPuns** stand out against other datasets: their 'negative' classes are not completely 'non-humorous'. The authors of the recently published TheNaughtyformer (**NF**) dataset (Tang et al., 2022) collected jokes, including dirty ones, from Reddit and complemented the humorous part with news articles. **NF** jokes are annotated as *clean*, *dark*, and *dirty* reflecting subreddits of their origin; we don't use this finer-grained annotation in our experiments.

Another group of datasets is based on human *editing*. West and Horvitz (2019) proposed the following method for building a dataset dubbed **Unfun.me**: they took satirical headlines from *The Onion* and asked volunteers to make them serious by minimal edits. In the current study we use the latest version of the dataset that is significantly larger than the initial edition, but retain only pairs with successfully 'unfunned' titles (based on subsequent annotation). Hossain et al. (2019) obtained Humicroedits dataset (**HME**) exploring an opposite direction: crowd workers had to modify a neutral news headline to obtain a funny one; the funniness of the modification was assessed on a later stage of the annotation. FunLines (**FL**) dataset was obtained in a similar fashion by volunteers, not through crowdsourcing (Hossain et al., 2020b). Since the datasets are very similar and haven't been used for humor detection task in their original form (hence we cannot compare our results with the previous art), we decided to merge HME and FL into one dataset. Additionally, we removed modified sentences with funniness score lower than 1 and

original headlines, for whom there are no funny modifications in the dataset. Editing-based datasets possess an appealing property: positive and negative examples are very similar on lexical and syntactic level, differing only in their funniness.

We complement the battery of humor datasets with a collection of headlines from *The Onion* (**TheO**). The motivation for including this data is to test the ability of the models to detect satire – a type of humor that refers to real-life events, as opposed to puns and other types of *linguistic* humor (**Unfun.me** is based on *The Onion* headlines, but is an order of magnitude smaller than **TheO**). Examples from all datasets can be found in the Table 7 in the Appendix.

## 3.2 Data processing

We divided the datasets, which don't have 'official' splits, into training, development, and test subsets in the ratio 70/10/20. Before splitting the data, we analyzed the data for duplicates within and across datasets. We used a straightforward approach to find repeated entries: exact string matching after lowercasing and removing punctuation. We excluded within-dataset duplicates (so dataset statistics may differ slightly from previously published figures) and ensured that all cross-dataset duplicates were in the training subsets to avoid data leakage. In the case of **ShJ**, **ReJ**, and **Haha**, we kept the original partitioning.

Motivated by the experiments on combined question answering datasets (Khashabi et al., 2020; Talmor and Berant, 2019) and our preliminary experiments, we also created a **COMB** dataset with the goal of diversifying joke sources. The dataset is a

combination of **PoD**, **Haha**, **Unfun.me**, and **TheO**. We also added 10,000 examples from the training and 5,000 from each of the validation and test sets of **ShJ**. We kept original train/dev/test splits of the constituents in the assembled dataset .

Table 1 summarizes the characteristics of the datasets in the study. As can be seen, the nine datasets vary greatly in size. We distinguish three size groups: *small* (up to 5k examples in both classes), *medium* (up to 50k), and *large*, represented by **NF** and **ShJ**. Not all datasets are balanced, and in some datasets the lengths of humorous and serious instances differ greatly (this is especially evident in the case of **NF**). Humor is often used to overcome taboos and express suppressed desires, that's why a considerable part of humor is 'dirty'. We counted dataset instances containing obscene words using a dedicated tool.[4] The proportion of texts containing obscene words is expectedly higher in **NF**, but the rest of the datasets (mainly based on Reddit content) are not completely 'pure' either. The KL-divergence shows how lexically similar the positive and negative classes are (lower values indicate greater lexical similarity of the two parts). As expected, the minimum values correspond to **FL+HME** and **Unfun.me**, where positive and negative examples differ by minimal edits (usually one-word substitutions). Both **ReJ** parts have the same source (Reddit), which explains the closeness of their word distributions. The creators of the **16kOL** and **PotD** datasets made special efforts when compiling the 'serious' parts of the datasets to minimize the lexical differences between the subsets, which is reflected in the low KL-divergence values. High divergence values suggest that many positive and negative examples can be separated on the basis of lexical features, which is confirmed by the results of the NB classifiers (see Table 8 in the Appendix).

### 3.3 Supplementary datasets

To test the behavior of humor detection models on out-of-domain data, we used several supplementary collections: sentences with figurative expressions, ironic tweets, dialogues from 19th century novels of different genres, and subtitles from TV series.

We leveraged the Fig-QA dataset (Liu et al., 2022), which is designed for Winograd-style evaluation and consists of sentence triples: a premise

with a metaphorical expression and two implications, one of which is correct. We applied the models to the premises and *correct* implications. We also tested the models on a collection of ironic tweets (Van Hee et al., 2018). Moreover, we collected two-turn dialogues from three 19th century English novels: *The Old Curiosity Shop* (1841) by Charles Dickens, *Alice's Adventures in Wonderland* (1865) by Lewis Carroll, and *Three Men in a Boat (To Say Nothing of the Dog)* (1889) by Jerome K. Jerome. The Dickens novel appears on several online lists of 'books that will make you cry', the Carroll novel is an example of the literary nonsense genre, is full of puns and parodies, while 'Three Men in a Boat' is a famous comic novel. We obtained the texts from Project Gutenberg and extracted all utterance pairs with no more than 200 characters between them. Similarly, we collected two-turn dialogs from two popular TV series: a horror *The Walking Dead* (WD) (2010-2022) and a sitcom *Friends* (1994-2004). We downloaded the subtitles from `OpenSubtitles.org` and formed pairs of utterances when they were no more than 10 seconds apart. These dialog collections can serve as approximation of real-life conversations, which we believe are the most promising application domain for humor detection models. The upper part of Table 3 summarizes the statistics of these additional datasets; examples can be seen in Table 9 in the Appendix.

Although the data has no low-level annotations, its genre can still provide useful insights. An expected behavior of a humor detection model is that it would fire more often on content from a comic novel than from a drama and on sitcom dialogues than on conversations from a horror. The results on these datasets can also provide insight into generalizability and robustness of the models, since the texts in this supplement are expected to be quite different from both the positive and negative examples of the humor datasets in the study.

## 4 Methods

### 4.1 Main classifiers

RoBERTa-base (Liu et al., 2019) is a workhorse in many NLP applications that has a good performance/size tradeoff. The model has 125M parameters, follows BERT's (Devlin et al., 2019) learning regime with some optimizations and slightly out-

---

[4] `https://github.com/snguyenthanh/better_profanity`

performs BERT on a number of tasks.[5] We used the RoBERTa implementation based on PyTorch from the HuggingFace library. We used a batch size of 4 for small and medium datasets, a batch size of 32 for large datasets, and a learning rate of $5 \times 10^{-5}$. We didn't use warmup for the learning rate when training on small datasets, and on the rest we used warmup for the first 100 steps. An early stop was used if there was no positive change over five evaluations steps on the development data. We used the Adam optimizer with a default weight decay of 0.01. Each model was trained for one epoch on the Nvidia Tesla V100 GPU. To ensure reproducibility of results, we used a fixed seed for the PyTorch, NumPy, and Python random number generators while training the models. Furthermore, we included a CUDA parameter that stipulated the use of deterministic algorithms.[6] Each model was trained with five different random seeds to obtain more accurate classification metrics (Dodge et al., 2020); we report median $F1$ scores. We also report results of smoothed multinomial NB classifiers.[7]

## 4.2 Third-party classifiers

We included two available third-party humor detection models into our experiments: a BERT-based humor classifier by Weller and Seppi (2019) (hereafter W&S) and ColBERT (Annamoradnejad and Zoghi, 2020).[8] W&S implements a standard architecture for BERT-based binary text classification. We reproduced the model using training scripts and **ReJ** datset published by the authors.[9] Official BERT-large checkpoint was trained for a single epoch with a learning rate of $2 \times 10^{-5}$ and a max sequence length of 128. ColBERT aims to model the setup-punchline structure of jokes and uses BERT-based embeddings of individual sentences as input. ColBERT was trained on a dataset of 100K humorous instances from ShortJokes and an equal number of news headlines. We use model weights published by the authors.[10]

## 4.3 LLMs as zero-shot classifiers

We complemented our experiments with two LLMs: Flan-UL2 (Tay et al., 2022) and chatGPT.[11] Flan-UL2 is an instruction-trained UL2 model with 20B parameters that slightly outperforms Flan-T5-XXL and approaches the performance of a larger Flan-PaLM on several benchmarks. Flan-UL2 is one of the few models that is affordable in terms of required computational resources with a non-restrictive license. ChatGPT is a model that was released by Open AI late 2022 and quickly gained worldwide popularity. Few details are revealed about chatGPT, it is described as a 'sibling' of InstructGPT (Ouyang et al., 2022). In our experiments, we use the `gpt-3.5-turbo-0301` version through its API. In both cases we used a simple zero-shot classification prompt.

In addition to classification, we also tasked the LLMs with joke continuation to test whether they had just memorized some of them during training (Carlini et al., 2023). To do this, we sampled 12,000 instances longer than one sentence from the **16kOL** (2,787) and **ReJ** (10,083). We fed the first sentence into the model and compared the returned continuation to the original with ROUGE-2 (Lin, 2004).

## 4.4 Adversarial attack

To test the robustness of humor detection models, we conducted a domain-specific adversarial attack. The goal of the attack was to deceive the models by substituting a single word in non-humorous texts. This attack can be seen as a simplified version of the humorous text generation proposed by Valitutti et al. (2013). Note that unlike attacking topic or sentiment classifiers, we don't need to preserve the original meaning of the text. The modification consists of the following steps:

1. The text is POS-tagged using spaCy;[12]
2. The last noun found in the text is sent to the Datamuse API, which returns up to 10 similarly sounding words;[13]
3. The word in the original sentence is replaced by the most distant word among the returned ones, based on the cosine similarity of the fast-Text embeddings (Bojanowski et al., 2017).

Due to limitations of the Datasmuse API, we sampled 500 examples from non-humorous parts of

---

[5]We conducted preliminary experiments also with RoBERTa-large. This resulted in marginally better results on large datasets, but very unstable behavior on small datasets (**PotD** and **EnPuns**), so we opted for the *base* version.

[6]See general reproducibility guidelines here: https://pytorch.org/docs/stable/notes/randomness.html

[7]https://scikit-learn.org/stable/modules/generated/sklearn.naive_bayes.MultinomialNB.html

[8]Not to be confused with an information retrieval model of the same name by Khattab and Zaharia (2020).

[9]https://github.com/orionw/RedditHumorDetection

[10]https://bit.ly/ColBERT

[11]https://openai.com/blog/chatgpt

[12]https://spacy.io

[13]https://www.datamuse.com/api/

each dataset. We examined a sample of modified text to confirm that the approach does not actually turn neutral text into jokes – at most, the modified text sounds nonsensical, for example: *Law catches flies, but lets **hornets** go free.* → *Law catches flies, but lets **hairnets** go free.*

## 5 Results and Analysis

### 5.1 Humor detection results

Table 2 summarizes obtained humor detection results. The diagonal elements of the upper part of the table show that the RoBERTa-based classifiers, trained and tested on the same datasets, achieve excellent results – in all but one cases the $F1$ values are above 0.9. The only exception is **ReJ** ($F1 = 0.70$), where both positive and negative classes consist of jokes. In most cases, the RoBERTa classifiers significantly outperform their NB counterparts, see Table 8. The gains are especially remarkable in case of the editing-based datasets – **Unfun.me** (+81 points) and **FL+HME** (+43 points). However, on three datasets – **NF**, **ShJ** and **EnPuns** – the gains are much smaller (in the case of **NF**, both the RoBERTa and NB classifiers achieve an almost perfect score a little less than 1.00). These are the datasets with largest lexical dissimilarity of humorous and non-humorous subsets (see KL-divergence scores in Table 1).

The transferability of the models varies in a wider range. The **PotD**-trained model shows worst results: on many datasets $F1$ is close to zero, the macro-averaged $F1$ is only 0.27. Note that **PotD** is the second smallest dataset in the study with 3,323 instances in the training set. **FL+HME** and **ReJ** also generalize poorly: macro-averaged $F1$ is slightly above 0.30. Interestingly, **Unfun.me**, the smallest dataset in the study, is quite competitive in terms of transferability, achieving a macro-averaged $F1 = 0.73$. The **Unfun.me** classifier trained on *The Onion* headlines and their 'unfunned' versions, achieves almost perfect recall (0.99) on a larger collection of the same origin. The **COMB**-based classifier outperforms other models in terms of macro-averaged $F1$ (0.80). This suggests that the diversity of the data contributes to a better transferability of humor detection. **ShJ** is not among the top results, indicating that there is no correlation between the dataset size and the outcome achieved. **NF** is the 'easiest' dataset of all. Somewhat surprisingly, despite its simplicity, **NF**-trained classifier generalizes quite well. De-

tecting satire in **TheO** also seems to be a relatively easy task for the RoBERTa-based models (although near-perfect recall scores may indicate overfitting). Both third-party classifiers show moderate results. ChatGPT and Flan-UL2 are very competitive: although they perform worse on individual datasets compared to the fine-tuned models, they are second and third best on many datasets and achieve quite high macro-averaged $F1$ scores.

We visualized dataset similarity in terms of classification transferability following (Talmor and Berant, 2019), see Figure 1. Each dataset is a node in a graph with edge weights defined as $\frac{F_1^{ij}}{F_1^{jj}} + \frac{F_1^{ji}}{F_1^{ii}}$, where $F_1^{ij}$ is the performance of RoBERTa-based classifier trained on $i$th dataset and tested on $j$th one. In the case of **TheO**, we use $2R^i$, double recall of the respective model. One can see that **FL+HME** and **PotD** are located on the periphery, while **COMB**, **Haha**, and **NF** are in the center. Notably, **COMB** and **Haha** are datasets containing humorous content from different sources. The centrality of **NF** is rather unexpected, given its rather straightforward organization.

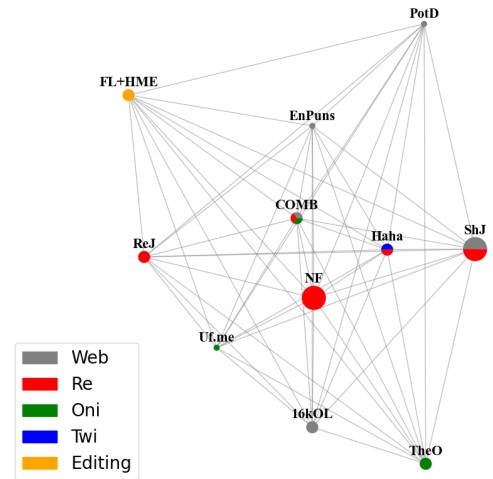

Figure 1: Dataset similarity based on classification results visualized using the *ForceAtlas2* algorithm. Colors indicate the source of humorous instances (see Table 1); node sizes reflect the dataset sizes.

### 5.2 Size of the training set

We trained a series of RoBERTa models on subsets of **ShJ** of increasing size and tested them on other datasets to find out how the amount of training affects the classification quality and transferability. As shown in Figure 2, only 0.5% of **ShJ** (2,330 examples) are enough to achieve a decent $F1$ score of 0.9 on the test subset of the same dataset. After

| | Model | 16kOL | PotD | EnPuns | ShJ | ReJ | Haha | FL+HME | Unfun.me | NF | COMB | avg $F1$ | TheO* |
|---|---|---|---|---|---|---|---|---|---|---|---|---|---|
| RoBERTa fine-tuned on... | 16kOL | **0.92** | 0.30 | 0.67 | 0.63 | 0.68 | 0.77 | 0.28 | 0.38 | 0.91 | 0.65 | 0.62 | 0.57 |
| | PotD | 0.01 | **0.98** | 0.09 | 0.64 | 0.04 | 0.23 | 0.12 | 0.01 | 0.09 | 0.52 | 0.27 | 0.01 |
| | EnPuns | 0.53 | 0.74 | **0.92** | 0.53 | 0.67 | 0.76 | 0.70 | 0.61 | 0.85 | 0.69 | 0.70 | 0.91 |
| | ShJ | 0.73 | 0.72 | 0.62 | **0.93** | 0.57 | 0.82 | 0.37 | 0.28 | 0.92 | 0.78 | 0.67 | 0.38 |
| | ReJ | 0.36 | 0.13 | 0.18 | 0.21 | **0.70** | 0.56 | 0.22 | 0.16 | 0.44 | 0.29 | 0.33 | 0.25 |
| | Haha | 0.62 | 0.71 | 0.70 | 0.84 | 0.66 | **0.94** | 0.36 | 0.40 | 0.95 | 0.79 | 0.70 | 0.47 |
| | FL+HME | 0.03 | 0.50 | 0.03 | 0.51 | 0.26 | 0.17 | **0.98** | 0.31 | 0.33 | 0.40 | 0.35 | 0.13 |
| | Uf.me | 0.68 | 0.54 | 0.79 | 0.58 | 0.66 | 0.74 | 0.72 | 0.92 | 0.92 | 0.73 | 0.73 | 0.99 |
| | NF | 0.67 | 0.68 | 0.85 | 0.65 | 0.67 | 0.76 | 0.72 | 0.61 | **1.00** | 0.75 | 0.74 | **1.00** |
| | COMB | 0.64 | 0.97 | 0.54 | 0.85 | 0.65 | 0.93 | 0.57 | **0.97** | 0.96 | **0.92** | **0.80** | 0.99 |
| | W&S | 0.49 | 0.52 | 0.53 | 0.28 | **0.70** | 0.45 | 0.41 | 0.46 | 0.21 | 0.40 | 0.44 | 0.46 |
| | ColBERT | 0.67 | 0.64 | 0.84 | 0.78 | 0.67 | 0.77 | 0.10 | 0.05 | 0.96 | 0.64 | 0.61 | 0.11 |
| | chatGPT | 0.77 | 0.85 | 0.78 | 0.85 | 0.67 | 0.91 | 0.42 | 0.56 | 0.70 | 0.82 | 0.73 | 0.59 |
| | FLAN | 0.75 | 0.74 | 0.66 | 0.86 | 0.67 | 0.91 | 0.65 | 0.69 | 0.96 | 0.83 | 0.77 | 0.67 |

Table 2: $F1$ scores of classifiers on individual datasets (* recall scores on **TheO** dataset) and macro-averaged $F1$ over all datsets (except **TheO**). Upper part: RoBERTa-based classifiers (median scores over five seeds), middle: two third-party models; bottom: 0-shot classification using LLMs. Best result for each dataset is in **bold**, second best is underlined.

| | | Fig-QA-p | Fig-QA-i | Irony | Alice | 3Men | Curiosity | Friends | WD | EB |
|---|---|---|---|---|---|---|---|---|---|---|
| | # instances | 9,106 | 9,106 | 1,911 | 579 | 230 | 2,004 | 8,392 | 9,791 | |
| | avg. length | 9.24 | 5.38 | 14.56 | 40.22 | 32.20 | 56.99 | 14.20 | 13.42 | |
| RoBERTa fine-tuned on... | 16kOL | 0.11 | 0.03 | 0.84 | 0.78 | 0.70 | 0.50 | 0.87 | 0.78 | + |
| | PotD | 0.08 | 0.08 | 0.13 | 0.00 | 0.01 | 0.00 | 0.02 | 0.03 | – |
| | EnPuns | 0.72 | 0.69 | 0.71 | 0.86 | 0.83 | 0.97 | 0.61 | 0.52 | – |
| | ShJ | 0.39 | 0.08 | 0.88 | 0.00 | 0.00 | 0.00 | 0.64 | 0.48 | – |
| | ReJ | 0.02 | 0.04 | 0.11 | 0.53 | 0.64 | 0.75 | 0.17 | 0.16 | – |
| | Haha | 0.57 | 0.21 | 0.23 | 0.60 | 0.66 | 0.42 | 0.61 | 0.50 | + |
| | FL+HME | 0.43 | 0.02 | 0.36 | 0.05 | 0.01 | 0.69 | 0.07 | 0.01 | – |
| | Uf.me | 0.97 | 0.99 | 0.78 | 0.96 | 1.00 | 0.93 | 0.79 | 0.97 | – |
| | NF | 1.00 | 1.00 | 1.00 | 1.00 | 1.00 | 1.00 | 1.00 | 1.00 | – |
| | COMB | 0.17 | 0.10 | 0.46 | 0.13 | 0.18 | 0.05 | 0.66 | 0.51 | + |
| | W&S | 0.33 | 0.31 | 0.31 | 0.28 | 0.27 | 0.40 | 0.21 | 0.24 | – |
| | ColBERT | 0.36 | 0.41 | 0.92 | 0.88 | 0.89 | 0.92 | 0.97 | 0.95 | – |
| | chatGPT | 0.51 | 0.07 | 0.51 | 0.67 | 0.49 | 0.45 | 0.40 | 0.16 | + |
| | FLAN | 0.53 | 0.03 | 0.69 | 0.45 | 0.33 | 0.22 | 0.43 | 0.22 | + |

Table 3: Statisitcs of the supplementary datasets (upper part) and proportion of instances classified as humorous by each of the models. Fig-QA-p – premises, Fig-QA-i – implications; average length is measured in words; EB (expected behavior): Fig-QA-p > Fig-QA-i, Alice > Curiosity, 3Men > Curiosity, Friends > WD.

that, the quality increases at a slower pace – gaining only 3 points when using the whole training subset. After an initial drop as the training set increases, the quality on **16OL** grows steadily as the training data increases. **16OL** is the only dataset that clearly benefits from increased training data. The models are relatively stable on **NF** and **COMB** throughout the experiments, although models trained on more than 3% of **ShJ** are inferior on **NF**. The plot also confirms that **NF** is a 'simple' dataset – training on about 1,165 positive and negative examples from **ShJ** only is sufficient to obtain $F1 = 0.95$. The performance of the models on **ReJ** decreases steadily with increasing training data, with a slight rebound for the model trained on the entire training set. This may be due to differences in the composition of **ShJ** and **ReJ**.

### 5.3 Robustness to the attack

Table 4 shows the percentage of cases where the adversarial attack deceived the classifier. As one can see, the **FL+HME**-based classifier is the most vulnerable – a simple attack affected almost a half of the labels. Note that the implemented modification resembles the human annotation scheme behind the dataset: a 'serious' news headline is turned into a funny one by minimal edits. At the same time, the **Unfun.me**-based classifier resists the attack quite well, even though the dataset construction is also based on human editing. We can assume

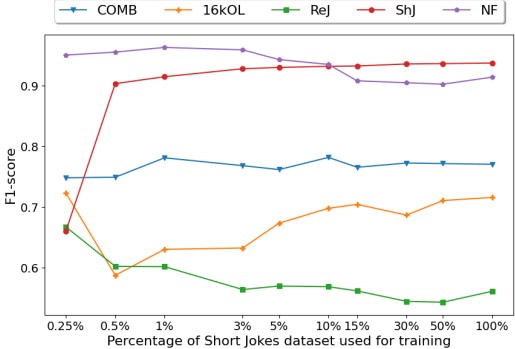

Figure 2: Impact of the training data size on the classification quality (median scores over five seeds).

that funny sentences in **Unfun.me** are funnier than those in **FL+HME**, so a dummy substitution in serious items doesn't make them look similar to their counterparts. Both models trained on pun collections – **EnPuns** and **PotD** – classified about 10% of the modified instances as funny. Presumably, the altered instances resemble homophone puns presented in the data. Models trained on **16kOL** and **ReJ** are least susceptible to the attack.

| Model | % of labels changed |
|---|---|
| 16kOL | 0.4 |
| PotD | 10.8 |
| EnPuns | 9.5 |
| ShJ | 3.5 |
| ReJ | 0.7 |
| Haha | 4.5 |
| FL+HME | 49.0 |
| Unfun.me | 2.3 |
| NF | 4.9 |
| COMB | 1.3 |

Table 4: Percentage of modified non-humorous instances classified as funny.

### 5.4 Behavior on supplementary datasets

The behavior of the models on supplementary collections is mixed, see Table 3. With some exceptions, the models assign texts to the humorous class more often than we expected, which may signal they didn't learn specific humor characteristics. **PotD** has detected very few humorous examples in the supplementary data, similarly to its behavior on the humor datasets; **ShJ** didn't trigger on fiction data at all. In contrast, **Unfun.me** and **NF** classify almost all instances as humorous. Some classifiers show a high recall on the collection of ironic tweets. Only five models don't violate our assumptions about a good humor detector on the supplementary data (regardless of the absolute val-

ues), see last column.

### 5.5 Memorization of jokes

The results in the Table 5 show that the humor recognition abilities of the LLMs are not based on their memorization of jokes during training. Although the ROUGE scores are generally low, we can see that chatGPT as a larger model was able to memorize more information, and an increased temperature parameter leads to more diverse responses. Out of a total of 12,000 examples, chatGPT reproduced 54 examples verbatim at $t = 0.2$ (e.g. *How do crazy people go through the forest? → They take the psychopath.*), but only 33 at $t = 1.5$ (the model continued the same prompt with *They go completely nuts!*).

| Model | 16kOL | ReJ | All |
|---|---|---|---|
| chatGPT (t = 0.2) | 0.038 | 0.022 | 0.025 |
| chatGPT (t = 1.5) | 0.028 | 0.014 | 0.017 |
| Flan-UL2 | 0.014 | 0.011 | 0.012 |

Table 5: Average ROUGE-2 scores of generated continuations.

## 6 Conclusions

Based on the results obtained, we can conclude that fine-tuned humor detection models perform well in laboratory settings. Their transferability depends more on the diversity of the training data than on its volume. The results of the models on out-of-domain data are unstable, suggesting that they are not yet mature enough to be used in practical applications. An adversarial attack showed that models trained on pun datasets are less robust. The results of LLMs on humorous datasets are quite competitive and don't rely on memorizing jokes, although they lag behind the results of the models trained specifically for humor detection. LLMs exhibit a more stable behavior on out-of-domain data, consistent with what is expected from a good humor detector. The results obtained on complementary datasets provide interesting insights and warrant further investigation.

In order to obtain a more realistic evaluation of humor detection models, we propose to include conversational data and to complement the evaluation with user studies. We believe that the obtained results will facilitate the development of new datasets and improved evaluation methodology in the field of computational humor.

## 7 Limitations

One limitation of our study is that we conducted experiments exclusively on English data. We only experimented with two types of trainable classifiers: RoBERTa- and NB-based. In the case of LLMs, we did not perform an exhaustive prompt tuning, but used a simple variant instead. In our work, we focus on the binary classification of humor and do not consider the related task of ranking or pairwise comparison of humorous texts. Furthermore, we only work with verbal humor, leaving outside the scope of the study multimodal humor, where the verbal component is combined with images or videos.

## 8 Ethics Statement

In our study, we use third-party datasets that may contain obscene words, may be offensive, and may not be politically correct. We provide a warning with the published data.

## Acknowledgements

We thank Rada Mihalcea and Diyi Yang for sharing the data, Grigory Sapunov and Leonid Boytsov for useful hints regarding LLMs, Pavel Efimov for providing us with scripts used for adversarial attacks. This research was supported in part through HPC facilities at HSE University (Kostenetskiy et al., 2021). Pavel Braslavski acknowledges funding from the School of Engineering and Digital Sciences, Nazarbayev University.

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

# A Complementary Materials

| Dataset | Download link | License |
|---------|---------------|---------|
| 16k one-liners (Mihalcea and Strapparava, 2005) | by request | – |
| Pun of the Day (Yang et al., 2015) | https://github.com/orionw/RedditHumorDetection | MIT |
| English Puns (Miller et al., 2017) | https://alt.qcri.org/semeval2017/task7/ | CC BY-NC |
| Short Jokes (Chen and Soo, 2018; Weller and Seppi, 2019) | https://github.com/amoudgl/short-jokes-dataset https://github.com/orionw/RedditHumorDetection | MIT / GPL-2.0 |
| Reddit Jokes (Weller and Seppi, 2019) | https://github.com/orionw/RedditHumorDetection | MIT |
| Unfun.me (West and Horvitz, 2019) | https://github.com/epfl-dlab/unfun | – |
| Humicroedits (Hossain et al., 2019) | https://cs.rochester.edu/u/nhossain/humicroedit.html | – |
| Funlines (Hossain et al., 2020b) | https://cs.rochester.edu/u/nhossain/funlines.html | – |
| Hahackathon (Meaney et al., 2021) | http://smash.inf.ed.ac.uk/hahackathon_data/ | – |
| The Naughtyformer (Tang et al., 2022) | https://github.com/leonardtang/The-Naughtyformer | – |
| TheOnion | https://github.com/lukefeilberg/onion | – |

Table 6: Datasets, bibliographic references, and download links.

| Dataset | ☺ | ☺ |
|---------|---|---|
| 16kOL | Couldn't afford to fix my brakes, so I made my horn louder. | Abbott says AIDS drug wins European approval. |
| PotD | in order to talk to a viking you need to know norse code | I believe that the people of this town want a mayor who s not afraid |
| EnPuns | Dateline London : Eccentric ornithologist travels to foreign land to teach pigeon English . | The best defense against logic is stupidity . |
| ShJ | Why couldn't the cop save the hippie from drowning? He was too far out man | If Hu is successful, he will be freer to boost spending on health, education and other services long-neglected in the headlong drive for economic growth. "The effect on the whole population needs to be considered, not just one age group." |
| ReJ | I just downloaded the Bohemian Rhapsody movie._____I think it was filmed in a movie theater, though - I see a little silhouetto of a man. | I went to the liquor store on my bicycle and bought a bottle vodka, put it in the basket on the front and then it occurred to me that if I fall or something happens, the bottle might break, so I drank it all right there and it's a good thing I did..._____...'cause I fell 7 times on the way home... |
| Haha | My wife thinks I don't give her enough privacy. At least that's what she said in her diary. | "If you love someone, you tell them. Even if you're scared that it's not the right thing. Even if you're scared that it'll cause problems." |
| FL+HME | Topless protesters crash pro-Franco bus in Madrid | Topless protesters crash pro-Franco demonstration in Madrid |
| Unfun.me | Obama's Declaration Of Swine Flu Emergency Prompts Pro-Swine-Flu Republican Response | Obama's declaration of swine flue emergency prompts pro-vaccine republican response |
| NF | My crush told me that she sees me as a brother I hope she's just as fond of incest as I am | Half of Japan firms see no escape from deflation this year: Reuters poll Around half of Japanese firms believe their country will have failed to rid itself of deflation a year from now, a Reuters poll shows, a sign that authorities are not gaining the traction they want as they battle an entrenched deflationary mindset... |
| TheO | North Korean Defector Says Kim Jong-Un Won't Last | – |

Table 7: Examples from humor datasets. In the case of **ReJ** a low line separates the title and body of the Reddit post; the negative example from **NF** is truncated.

| | Model | 16kOL | PotD | EnPuns | ShJ | ReJ | Haha | FL+HME | Unfun.me | NF | COMB | avg $F1$ | TheO* |
|---|---|---|---|---|---|---|---|---|---|---|---|---|---|
| Naïve Bayes trained on... | 16kOL | **0.81** | 0.61 | 0.76 | 0.79 | 0.66 | 0.77 | 0.40 | 0.43 | 0.97 | 0.75 | 0.69 | 0.61 |
| | PotD | 0.30 | 0.70 | 0.78 | 0.37 | 0.34 | 0.42 | 0.47 | 0.46 | 0.46 | 0.45 | 0.47 | 0.35 |
| | EnPuns | 0.52 | **0.74** | **0.89** | 0.56 | 0.62 | 0.71 | **0.64** | 0.55 | 0.79 | 0.66 | 0.67 | 0.71 |
| | ShJ | 0.71 | 0.60 | 0.74 | **0.89** | 0.66 | 0.83 | 0.39 | 0.29 | 0.98 | 0.77 | 0.69 | 0.44 |
| | ReJ | 0.37 | 0.32 | 0.46 | 0.24 | 0.52 | 0.36 | 0.35 | 0.30 | 0.31 | 0.31 | 0.35 | 0.25 |
| | Haha | 0.58 | 0.67 | 0.77 | 0.77 | 0.64 | **0.87** | 0.49 | 0.50 | 0.95 | 0.74 | 0.70 | 0.50 |
| | FL+HME | 0.63 | 0.64 | 0.75 | 0.67 | 0.61 | 0.77 | 0.55 | **0.56** | 0.86 | 0.70 | 0.67 | 0.64 |
| | Uf.me | 0.57 | 0.52 | 0.63 | 0.59 | 0.62 | 0.64 | 0.38 | 0.10 | 0.73 | 0.59 | 0.54 | 0.53 |
| | NF | 0.73 | 0.65 | 0.80 | 0.83 | **0.67** | 0.78 | 0.33 | 0.52 | **1.00** | 0.78 | 0.71 | 0.64 |
| | COMB | 0.67 | 0.68 | 0.79 | 0.86 | 0.66 | **0.87** | 0.63 | 0.45 | 0.98 | **0.84** | **0.74** | **0.80** |

Table 8: $F1$ scores of Naïve Bayes classifiers on individual datasets (* recall scores on **TheO** dataset) and macro-averaged $F1$ over all datsets (except **TheO**); best result for each dataset is in **bold**, second best is underlined.

| Dataset | Example | # |
|---|---|---|
| Fig-QA premises | when the food arrived it was as hot as ice cream | 9 |
| | the babysitter was as mature as Mary Poppins | 9 |
| | That 400-pound man needs another donut as much as a baby needs scotch whiskey. | 7 |
| | The microphone is as jumpy as mud | 6 |
| Fig-QA implications | The food was cold | 4 |
| | the sitter was competent | 6 |
| | That 400-pound man does not need another donut. | 9 |
| | the microphone is highly insensitive / underpowered. | 9 |
| Irony | oh lord! RT @popularmsem: RT @ShockingFactsz: Before becoming an actor, Tom Cruise wanted to be a Catholic priest. | 10 |
| | This is the weather i just love walking to work in #worst #weatherbomb | 9 |
| Alice | "Come, let's try the first figure!" said the Mock Turtle to the Gryphon. "We can do without lobsters, you know. Which shall sing?" "Oh, _you_ sing," said the Gryphon. "I've forgotten the words." | 8 |
| | "How should _I_ know?" said Alice, surprised at her own courage. "It's no business of _mine_." "Off with her head! Off—" | 8 |
| 3Men | "It's very dark. Why don't you light the gas?" "Oh!" | 8 |
| | "How could I wake you, when you didn't wake me?" he retorted. "Now we shan't get on the water till after twelve. I wonder you take the trouble to get up at all." "Um," I replied, "lucky for you that I do. If I hadn't woke you, you'd have lain there for the whole fortnight." | 8 |
| Curiousity | 'To dinner!' thought Dick, 'that's another circumstance. I don't believe that small servant ever has anything to eat.' 'Sammy won't be home,' said Miss Brass. 'Stop till I come back. I sha'n't be long.' | 9 |
| | 'A very little one,' 'Miss Sally couldn't kill me if she know'd I went down there, so I'll come,' said Richard, putting the cards into his pocket. 'Why, how thin you are! What do you mean by it?' | 9 |
| Friends | l can't believe you haven't told that girl she doesn't have a job. You haven't taken down the Christmas lights. | 10 |
| | I foId, Iike a cheap hooker who got hit in the stomach by a fat guy with sores on his face. | 10 |
| WD | My wife would lock the bedroom door. I'm sad to say that couch and I become old friends. | 10 |
| | Working on your tan with a shotgun in your lap. No, I am on watch against walkers. | 10 |

Table 9: The funniest examples from supplementary datasets according to the majority of 10 RoBERTa-based classifiers: # – number of models classified the item as funny (in case of Fig-QA datset we cite the 'funniest' premises along with their correct implications and vice versa). Note that in the second *Friends* example two words contain capital 'i' instead of 'l' – seemingly due to OCR errors.