# OpenReview forum: "You Told Me That Joke Twice: A Systematic Investigation of Transferability and Robustness of Humor Detection Models"
_EMNLP/2023/Conference — EMNLP 2023 Conference Withdrawn Submission_

### Official Review · Reviewer_3afp · 2023-08-04

**Soundness:** 4

**Excitement:**

4: Strong: This paper deepens the understanding of some phenomenon or lowers the barriers to an existing research direction.

**Missing References:**

The following work does some experiments that are very similar to the robustness experiments, to create a negative set of non-jokes using the same vocabulary as jokes:
- https://arxiv.org/abs/2010.13652

Some more references to (computational) humor theory:
- https://direct.mit.edu/books/book/3352/Inside-JokesUsing-Humor-to-Reverse-Engineer-the
- Computers Learning Humor Is No Joke: T Winters - Harvard Data Science Review, 2021

**Paper Topic And Main Contributions:**

This paper presents a dataset on humor detection and the authors investigate the performance of different classifiers on this dataset, as well as on other datasets. In addition, they perform an extra study of the robustness of classifiers to jokes with homonyms.

**Reasons To Accept:**

I think the paper is interesting and the contribution of a dataset (and standard splits for other datasets) might be useful to other practitioners. I appreciated the dataset descriptions and examples (mostly in appendices), as well as the details on the model training.

The experiments are interesting and evaluating zero-shot LLMs as well is definitely useful, since this gives a sense of their capacity of "understanding" jokes.

**Reasons To Reject:**

The current manuscript is reasonably well-written and provides some nice experiments, aside from contributing a dataset. However, I am missing some more depth.

One direction could be to focus more on the linguistic or theoretical aspects of humor. The current paper dives a bit into this with the robustness experiment, but even there it stays quite on the surface. The introduction is a bit hand-wavey as well, perhaps a more grounded paragraph (e.g. linking to works about incongruity theory) would improve this.

Another direction could be to flesh out the experiments a bit more. Cluster the datasets in groups, so that we can learn some more higher-level trends other than "model Y performs poorly on dataset X." Which datasets are similar and which are not? Which results surprise us?

I am also mostly missing a conclusion on the experimental results. The current manuscript just provides a table with a lot of numbers (this could be presented better, perhaps with colors?) and a textual description of the results, yet no real interpretation. Adding such interpretation would make the paper a lot more interesting as a reader.

**Reproducibility:**

4: Could mostly reproduce the results, but there may be some variation because of sample variance or minor variations in their interpretation of the protocol or method.

**Reviewer Confidence:**

4: Quite sure. I tried to check the important points carefully. It's unlikely, though conceivable, that I missed something that should affect my ratings.

**Typos Grammar Style And Presentation Improvements:**

- Perhaps mention somewhere that COMB is your dataset? This contribution can be better highlighted.

---

> ### Author Rebuttal · Authors · 2023-08-28
>
> Thank you for your time and thoughtful comments! Please find our response below.
>
> We do not directly use **theories of humor** in our main experiments (in contrast to earlier humor detection studies that used humor feature engineering). However, we do use ideas from incongruity theory when generating data for adversarial attacks. We will make this connection more explicit and describe the theories that have been used in previous works on computational humor in the introduction.
>
> **Clustering of datasets.** We analyzed the datasets along many dimensions, including source of funny and non-funny texts, size, and lexical proximity of positive and negative classes. In addition, we formalized the proximity of the datasets based on the success of transfer learning; these results are presented in the appendix (Fig. 2). If the paper is accepted, we will be able to move these results to the main content thanks to the additional space.
>
> Thank you very much for the very relevant **references**, we'll include them in the next edition of the paper.
>
> **Representation of results and conclusion.** In case the paper is accepted we will use extra space to elaborate on interpretation of the results. In particular, we will discuss the most unexpected result -- the size of the dataset is not its main advantage. We experimented with color presentation of the main experimental results, but opted for black and white for the convenience of color-blind readers.
>
> The **COMB dataset** is not quite ours – it was compiled from existing humor datasets with the goal of diversifying and balancing humor types and text sources. We will improve the description of the dataset to make it clearer.

---

### Official Review · Reviewer_ff1i · 2023-08-05

**Soundness:** 5

**Excitement:**

4: Strong: This paper deepens the understanding of some phenomenon or lowers the barriers to an existing research direction.

**Paper Topic And Main Contributions:**

In this paper, the authors evaluated the impact of large language models (Flan-U2 and chatGPT) and the robustness of humor detection models.
The authors present their results on nine datasets, i.e., 16kOL, PotD, EnPuns, ShJ, ReJ, Haha, FL+HME, Unfun.me, NF, TheO, and COMB.

This paper has a clear contribution to humor detection and will certainly have a relevant impact on future research.  This paper conducts a range of experiments studying the proposed objectives.

**Reasons To Accept:**

Strength:

The authors standardize previously available collections for humor detection, thus making it possible to compare the different methods.

The authors make available the standardized collection.

The paper contains a good level of implementation details.

Extensive experiments are conducted, and multiple baselines are provided.

**Reasons To Reject:**

Weaknesses:

The authors only experimented with two types of classifiers, namely RoBERTa and Naïve Bayes.

The authors only experimented with two large language models, namely Flan-U2 and chatGPT.

**Reproducibility:**

4: Could mostly reproduce the results, but there may be some variation because of sample variance or minor variations in their interpretation of the protocol or method.

**Reviewer Confidence:**

3: Pretty sure, but there's a chance I missed something. Although I have a good feel for this area in general, I did not carefully check the paper's details, e.g., the math, experimental design, or novelty.

---

> ### Author Rebuttal · Authors · 2023-08-28
>
> Thank you for your time, insightful comments and questions! Please find our response below.
>
> Our goal was to evaluate humor detection and its transferability using Transformer-based models that have demonstrated SotA performance in recent studies. As shown by Peyrard et al, 2021, cognate models (BERT, distilBERT, RoBERTa) yield similar results, so we opted for the RoBERTa model. The addition of Naïve Bayes classifiers allowed us to assess the progress made in the field since the pioneering work of Mihalcea & Strapparava (2005). At the time we started the study, only one LLM (ChatGPT) was available through an API. At the same time, our computing resources allowed us to host only Flan-U2, while the status of the leaked LLaMa was debatable. We plan to include the GPT4 results in the next edition of the paper.

---

### Official Review · Reviewer_JhfG · 2023-08-05

**Soundness:** 4

**Excitement:**

3: Ambivalent: It has merits (e.g., it reports state-of-the-art results, the idea is nice), but there are key weaknesses (e.g., it describes incremental work), and it can significantly benefit from another round of revision. However, I won't object to accepting it if my co-reviewers champion it.

**Paper Topic And Main Contributions:**

This paper focuses on the task of verbal humor detection and investigate the existing English humor datasets and how various models behave and generalize on these datasets. Before training the models, the authors collected nine relevant datasets that appear in previous literature and added one more by themselves, to test the models with out-of-domain data. They also made a combined dataset out of these existing ones in order to diversify the humorous texts, thus improving the transferability of the models. They compared RoBERTa-based models with existing 3rd-party classifiers such as ColBERT, as well as LLMs in zero-shot scenarios. Experimental results showed that in most cases RoBERTa-based models perform the best when trained and tested on the same data, and better generalization was achieved on the combined dataset, with various types of humorous texts taken from each dataset. On the other hand, LLMs generally gave competitive results across different datasets, and didn't achieve this through memorization of jokes. The authors argue that the obtained results could facilitate the development of computational humor, and all the datasets used in the experiments are made publicly available.

**Questions For The Authors:**

Since you mentioned that humor detection is promising for conversational systems, I am wondering if any of the datasets you used contain text in dialog form? And in your opinion, how is the task of humor detection in dialogs different from that in jokes? What aspects do you plan to consider (data, model, evaluation, etc.)?

**Reasons To Accept:**

The main reasons to accept this paper are:
- The paper provides a comprehensive investigation of the existing verbal humor detection datasets (in English), without missing ones as far as I know, and shows how current NLP models perform on these datasets;
- The paper gives interesting insight on the models' performance with respect to the KL divergence of the word distributions in positive and negative classes of the training data, which could imply that the trained model actually predict results based on words, instead of having a real sense of humor;
- The datasets used in the paper are cleaned and unified, and are made publicly available.

**Reasons To Reject:**

The main reasons to reject this paper are:
- Despite the fact that the paper investigate the performance of NLP models on existing verbal humor datasets, the main results seem trivial (e.g., fine-tuned models perform the best on their respective datasets);
- The extra dataset curated by the authors does not have annotations, thus having limited usage for humor detection tasks;
- It would be good if experiments on the supplementary dataset include some human evaluation;
- Although the authors mentioned humor detection in conversational scenarios (both in introduction and conclusion), it is not clear how the bulk of the paper content (dataset and models) is connected to conversations.

**Reproducibility:**

5: Could easily reproduce the results.

**Reviewer Confidence:**

5: Positive that my evaluation is correct. I read the paper very carefully and I am very familiar with related work.

---

> ### Author Rebuttal · Authors · 2023-08-28
>
> Thank you for your time and valuable comments! Please find our responses below.
>
> **Trivial results.** Indeed, it is an expected result that models perform best on the datasets on which they were trained. However, we believe that simultaneous experiments on multiple datasets using different models yield much more interesting and versatile results: transferability varies across datasets and models, in some cases fine-tuned Transformer models do not outperform Naïve Bayes classifiers, the fine-tuned models behave quite differently on data from fiction books and TV series, the size of a dataset is not a decisive factor, etc.
>
> **Utility of data without annotations.** We think that the collected data can provide some insights even without annotation (although genre and domain are already a kind of high-level annotation): the data helps to evaluate the robustness and signals potential overfitting of the models. Currently, we don't have resources to partially annotate the data, but we plan to do so as future work.
>
> **Conversational data for humor detection.** We believe that conversational systems are the most promising application for humor detection. At the same time, existing humor datasets contain very little conversational data. They may contain two-turn jokes and social media texts that resemble dialogues, but this similarity is rather superficial -- jokes in dialogues are more spontaneous and context-dependent. Dialogues from TV series and fiction books can be seen as approximations of conversational data (Henderson, Budzianowski et al., 2019). In our study, we show that humor detection models tend to 'overreact' to this data. We hypothesize that this is because the 'serious' parts of the datasets (usually news) are quite different from colloquial speech. In the future, we plan to work on a methodology for evaluating humor detection that combines conversational data and user studies.

---

### Meta-Review · Area_Chair_iTfT · 2023-09-17

**Recommendation:** 5

**Metareview:**

The paper delves into verbal humor detection, examining English humor datasets and the performance of various models on them. Nine datasets from previous literature were collected, and an additional one was created by the authors for out-of-domain data testing. These datasets were combined to diversify humorous texts, enhancing model transferability. RoBERTa-based models were compared with 3rd-party classifiers like ColBERT and large language models (LLMs) in zero-shot scenarios. Results indicated that RoBERTa models excel when trained and tested on identical data, especially on the combined dataset. LLMs, however, showed competitive results across datasets without merely memorizing jokes. The authors believe their findings can advance computational humor and have made all datasets publicly accessible.

The paper offers a thorough exploration of English verbal humor detection datasets and evaluates current NLP model performances on them. It provides valuable insights into model performance based on word distributions in training data, suggesting models might predict based on words rather than genuine humor understanding. The datasets used have been cleaned, unified, and made publicly available. The authors have standardized prior humor detection collections, enabling method comparison, and have shared this standardized collection. The paper is detailed in its implementation and conducts extensive experiments with multiple baselines. The contribution of a new dataset and standardized splits for existing ones is seen as beneficial for practitioners. The paper's evaluation of zero-shot LLMs is also deemed valuable for understanding their joke comprehension capabilities.

The main weaknesses have been discussed during the rebuttal phase and are listed below.
The paper, while investigating NLP model performance on humor datasets, presents seemingly obvious results, such as fine-tuned models performing best on their specific datasets. The additional dataset curated by the authors lacks annotations, limiting its utility for humor detection. There's a disconnect between the paper's content and its mention of humor detection in conversational scenarios. The research is limited in its experimentation, focusing only on two classifiers (RoBERTa and Naïve Bayes) and two LLMs (Flan-U2 and chatGPT). The paper could benefit from a deeper exploration into the linguistic or theoretical facets of humor. The introduction lacks depth, and the experiments could be more detailed. The manuscript's presentation of results is mainly numerical without a comprehensive interpretation, making it less engaging for readers.

---

### Decision · Program_Chairs · 2023-10-07

**Decision:**

Accept-Main

**Comment:**

The paper delves into verbal humor detection, examining English humor datasets and the performance of various models on them. Nine datasets from previous literature were collected, and an additional one was created by the authors for out-of-domain data testing. These datasets were combined to diversify humorous texts, enhancing model transferability. RoBERTa-based models were compared with 3rd-party classifiers like ColBERT and large language models (LLMs) in zero-shot scenarios. Results indicated that RoBERTa models excel when trained and tested on identical data, especially on the combined dataset. LLMs, however, showed competitive results across datasets without merely memorizing jokes. The authors believe their findings can advance computational humor and have made all datasets publicly accessible.

The paper offers a thorough exploration of English verbal humor detection datasets and evaluates current NLP model performances on them. It provides valuable insights into model performance based on word distributions in training data, suggesting models might predict based on words rather than genuine humor understanding. The datasets used have been cleaned, unified, and made publicly available. The authors have standardized prior humor detection collections, enabling method comparison, and have shared this standardized collection. The paper is detailed in its implementation and conducts extensive experiments with multiple baselines. The contribution of a new dataset and standardized splits for existing ones is seen as beneficial for practitioners. The paper's evaluation of zero-shot LLMs is also deemed valuable for understanding their joke comprehension capabilities.

The main weaknesses have been discussed during the rebuttal phase and are listed below.
The paper, while investigating NLP model performance on humor datasets, presents seemingly obvious results, such as fine-tuned models performing best on their specific datasets. The additional dataset curated by the authors lacks annotations, limiting its utility for humor detection. There's a disconnect between the paper's content and its mention of humor detection in conversational scenarios. The research is limited in its experimentation, focusing only on two classifiers (RoBERTa and Naïve Bayes) and two LLMs (Flan-U2 and chatGPT). The paper could benefit from a deeper exploration into the linguistic or theoretical facets of humor. The introduction lacks depth, and the experiments could be more detailed. The manuscript's presentation of results is mainly numerical without a comprehensive interpretation, making it less engaging for readers.